# The Impact of COVID-19 on Eating Environments and Activity in Early Childhood Education and Care in Alberta, Canada: A Cross-Sectional Study

**DOI:** 10.3390/nu13124247

**Published:** 2021-11-26

**Authors:** Lynne M. Z. Lafave, Alexis D. Webster, Ceilidh McConnell, Nadine Van Wyk, Mark R. Lafave

**Affiliations:** Department of Health and Physical Education, Mount Royal University, Calgary, AB T3E 6K6, Canada; adwebster@mtroyal.ca (A.D.W.); ckmcconnell@mtroyal.ca (C.M.); nvanwyk@mtroyal.ca (N.V.W.); mlafave@mtroyal.ca (M.R.L.)

**Keywords:** healthy eating, nutrition, COVID-19, early childhood education and care, preschooler, childcare, accelerometer, physical activity, MVPA, sedentary

## Abstract

Early childhood education and care (ECEC) environments influence children’s early development and habits that track across a lifespan. The purpose of this study was to explore the impact of COVID-19 government-mandated guidelines on physical activity (PA) and eating environments in ECEC settings. This cross-sectional study involved the recruitment of 19 ECEC centers pre-COVID (2019) and 15 ECEC centers during COVID (2020) in Alberta, Canada (*n* = 34 ECEC centers; *n* = 83 educators; *n* = 361 preschoolers). Educators completed the CHEERS (Creating Healthy Eating and activity Environments Survey) and MEQ (Mindful Eating Questionnaire) self-audit tools while GT3X+ ActiGraph accelerometers measured preschooler PA. The CHEERS healthy eating environment subscale was greater during COVID-19 (5.97 ± 0.52; 5.80 ± 0.62; *p* = 0.02) and the overall score positively correlated with the MEQ score (*r* = 0.20; *p* = 0.002). Preschoolers exhibited greater hourly step counts (800 ± 189; 649 ± 185), moderate-to-vigorous PA (MVPA) (9.3 ± 3.0 min/h; 7.9 ± 3.2 min/h) and lower sedentary times (42.4 ± 3.9 min/h; 44.1 ± 4.9 min/h) during COVID-19 compared to pre-COVID, respectively (*p* < 0.05). These findings suggest the eating environment and indices of child physical activity were better in 2020, which could possibly be attributed to a change in government-mandated COVID-19 guideline policy.

## 1. Introduction

The coronavirus disease (COVID-19) outbreak was declared a global pandemic on 11 March 2020 by the WHO and early childhood service programs in Alberta shut down on 16 March 2020 [1,2]. One month after the initial shut down, early childhood education and care (ECEC) centers in Alberta were allowed to re-open under new regulated guidelines implemented minimize the spread of the virus [3]. General guidelines in Alberta, Canada included limitations of in-person visitation, stringent surface sanitization, and the establishment of cohorts to reduce contact with colleagues and other families. Food-related guidelines prohibited self-serve or family style meal services, as well as common foods, and, where possible, enforced physical distancing while eating. Physical activity-related guidelines included the sanitization of play structures between cohort use, the designation of play equipment to a single cohort, and the use of alternative spaces for physical activity such as community walks, supervised play in local parks and safe open spaces, or public playgrounds with children engaging in hand washing before and after play (August 2020) [3]. These new guidelines altered the usual functioning of center-based childcare.

Early childhood is a critical period of cognitive, social, and emotional growth where health-related habits established during this time tend to persist throughout the lifespan [4,5,6]. In Canada, approximately 60% of children aged 0–5 years participate in some type of childcare arrangement with over half of these attending center-based care [7]. Of those who attend childcare arrangements, a large proportion of their waking time (6 h on average) is spent in care [8]. This suggests that structured care settings play a pivotal role in shaping the future health and well-being behaviours of future adults.

Supporting children that attend childcare arrangements to become healthy eaters and physically active individuals is a collaborative responsibility for educators and parents. Healthy eating behaviours that support the intake and choice of high-nutritional-value foods support child growth and development [9]. The development of these behaviours can take a variety of approaches to improve markers of healthy eating such as increased vegetable and fruit intake [10]. Quality nutrition intake is positively associated with motor development and cognitive functioning in infancy, preschool years, adolescence, and early adulthood, as well as an associated reduced risk of common non-communicable diseases such as obesity, diabetes, and cardiovascular disease [11,12,13,14].

Similarly, physical activity supports children’s health by enhancing emotional, cognitive, and physical health [15,16]. The “24-h” movement behavior guidelines for children less than 5 years of age, developed in 2017 in Canada, Australia, and New Zealand and again by the WHO in 2019, provide clear direction and support on physical activity targets for caregivers of young children [17,18]. Specifically, total physical activity (TPA) and moderate-to-vigorous physical activity (MVPA) for preschool-aged children are positively associated with favorable health indicators such as motor development, cognitive development, psychosocial health, bone and skeletal health, and cardiometabolic health [16,19].

The purpose of this cross-sectional study was to investigate the impact of COVID-19 government-mandated guidelines on eating and physical activity environments in early childhood education and care (ECEC) centers. A comprehensive understanding of changes in center practices in response to the pandemic could be useful to inform future public health interventions. Tool development is iterative and ongoing. A secondary purpose was to evaluate the relationship between CHEERS and MEQ to continue the investigation of construct and concurrent validity.

## 2. Materials and Methods

### 2.1. Participants and Design

This cross-sectional study involved 34 ECEC centers, with data collected pre-COVID in the fall of 2019 (Y-19) and during COVID-19 in the fall of 2020 (Y-20), across the province of Alberta, Canada as part of a larger study investigating a health and wellness educational support program. Eligibility requirements for childcare center recruitment in the fall of 2019 and 2020 were matched to ensure the recruited centers were from similar geographic locations and cities with similarly sized centers, and had similar auspices (profit/not-for-profit) and preschooler-care characteristics, in order to guarantee the homogeneity of the sample [20,21]. The larger study served as the sampling frame for this study, with baseline data from the trial presented within this paper. Ethical approval to conduct the study was obtained from the Mount Royal University Human Research Ethics Board (no. 101768).

ECEC centers from two metropolitan and two mid-sized cities across Alberta, Canada were invited to participate in the study. Recruitment took place between July and August in 2019 for Y-19 (pre-pandemic) and again in July and August 2020 for Y-20 (during the pandemic). Questionnaire, anthropometric, and accelerometer data were collected during a seven-week period between September and October in 2019 and 2020, respectively.

#### 2.1.1. Childcare Centers and Educators

The target population for the study was licensed ECEC centers identified as Day Care Programs (Schedule 1 of the Childcare Licensing Regulation) that are facility-based centers that serve infants, toddlers, and preschool-aged children. They typically provide care throughout the day, from the morning to early evening. For ECEC centers to be eligible for the study, they had to (1) provide care for a minimum of 15 preschool-aged (3–5 years) children with the classification of day care program, as opposed to family day home or after school care programs; (2) have internet access for the center; and (3) not be currently participating in any other intervention to improve healthy eating and/or physical activity. For educators to be eligible for the study, they had to be working full time at the ECEC center and assigned to a preschool-aged room.

ECEC centers were randomly selected for recruitment using postal codes to stratify the selection of ECEC centers from large urban population centers (population > 100,000), medium population centers (30,000–99,000), small population centers (1000–29,999), and rural areas (population < 1000) throughout the province [22]. In 2019, center directors were contacted by phone, provided with a brief summary of the research, and invited to participate in the study. Directors invited educators to participate. Those agreeing to participate received an email package with instructions, a consent form, links to surveys, and contact information of a trained research associate to answer potential questions. In 2020, a new cohort of centers were contacted to participate in the study following the same protocol.

Participating ECEC staff completed demographic questions that comprised items such as sex, age, highest level of education completed, type of childcare facility, and length of time as an educator. Population center categories were determined from postal code data.

#### 2.1.2. Children

For children to be eligible, they were required to: (1) have prior written consent from a parent or guardian; (2) be aged between 3 and 5 years; and (3) be enrolled full time at the center. Once the ECEC centers had consented to participate in the study, center staff were asked to distribute informational flyers and consent forms to parents with children aged between 19 and 70 months via center communication methods with parents. For those centers that consisted of more than one classroom, staff were asked to only distribute information statements and consent forms to the classroom with the highest number of children enrolled aged between 2 and 5 years. Prior to the data collection and signing of the consent forms, parents and educators were given written and oral information about the procedures and ethical standards for testing. Flyers provided to parents had phone numbers of research staff and paper consent forms with details of the study. Among other things, these included information about the participants’ right to decline to participate, and their right to withdraw from the research once it had started. Written consent forms from guardians were collected by research staff for participation and children provided verbal assent during data collection.

Parents answered demographic questions related to birth date and sex when returning the consent forms. Anthropometric data were collected in the morning of the first day of accelerometer data collection in the childcare center. Classroom educators were trained on anthropometric data collection by the research team prior to data collection periods. A team of two classroom educators measured children’s standing height to the nearest millimeter using a portable stadiometer (Model 213; Seca, Hamburg, Germany) and a digital scale to collect body weight to the nearest 0.1 kg. Height, weight, sex, date of birth, and date of data collection were used to calculate body mass index (BMI) for age [23].

### 2.2. Questionnaires

*Creating Healthy Eating and activity Environments Survey (CHEERS) tool.* The creating healthy eating and active environments survey (CHEERS) tool is a community-based, educator-administered self-audit tool designed to offer ECEC centers an evaluative measure for eating and activity environments for their childcare context. This tool has been assessed for reliability and validity with early childhood experts and educators [24,25]. The tool is completed by an ECEC staff member and provides a score for the overall assessment and four subscale scores for the center. CHEERS includes 59 items with four proposed subscales: food served (*n* = 23), healthy eating environment (*n* = 18), healthy eating program planning (*n* = 6), and physical activity environment (*n* = 12). The CHEERS score is calculated by a cumulative average of the four subscales (score range 4–28). The four subscale scores are calculated using an average of the items in the grouping. Of the 59 items, 86% are scored from 1 = “never” to 7 = “always”. The response options for policy items are specific to availability relative to child enrollment in the ECEC center (week–year). A higher score indicates alignment with best practices in nutrition and physical activity [25]. ECEC staff completed the CHEERS survey directly through an online portal in the Qualtrics survey platform.

*Mindful Eating Questionnaire (MEQ).* The MEQ is a valid and reliable self-report instrument to assess mindful eating in healthy adults [26]. It consists of 28 items that measure five domains of mindful eating: disinhibition (*n* = 8)—ability to stop eating when full; awareness (*n* = 7)—attentiveness to food characteristics (texture, smell, and taste); external cues (*n* = 6)—propensity to eat in response to external cues; emotional response (*n* = 4)—tendency to eat in response to negative emotions; and distraction (*n* = 3)—level of distraction while eating [26]. The emotional and distraction subscales are reverse scored and 5 questions on disinhibition are reverse scored. Each item is scored from 1 = “never/rarely” to 4 = “usually/always”, where higher scores signify more mindful eating. Each subscale score is calculated as the mean of the items within it, excluding those with a “not applicable” response. The summary score is the mean of the five subscales.

### 2.3. Accelerometry

ActiGraph GT3X+ was developed to establish validity and reliability in the assessment of objective physical activity (PA) levels in preschool-aged children in order to provide data on physical activity counts, energy expenditure, steps taken, and intensity levels [27,28,29,30]. ActiGraph GT3X+ accelerometers (ActiGraph. Pensacola, FL, USA; sampling frequency = 30 Hz), which measure the three-dimensional acceleration of the body, were collected in 15 s epochs and sedentary time, light-intensity PA (LPA), and MVPA were classified using validated cut points [29]. To adjust for variation in wear time, average steps per hour and average minutes per hour, for each physical activity intensity level (sedentary, LPA, MVPA, TPA), were calculated [31]. Accelerometer data were also categorized into the percent of the day spent sedentary, in LPA, in MVPA, and in any physical activity (TPA).

Participating 3–5-year-olds wore the accelerometers, attached to an elastic belt and tightly fixed above the right hip, for seven childcare days. Classroom educators were trained on correct placement and removal of accelerometers. Children were fitted when they arrived at their childcare center and accelerometers were removed at the end of each day. Activity logs were used to record placement and removal times as well as nap times. Educators were instructed to support children wearing the monitor for the full day of activities.

The raw data were downloaded into ActiLife-software, version 6.13.4 (ActiGraph, Pensacola, FL, USA) for analysis. Non-wear time was defined as periods of consecutive zero activity counts of ≥60 min [32]. Participant data were included in the analysis if wear time validation indicated the participant wore the accelerometer for at least 250 min per day for a minimum of four days. Wear time was also validated through cross examination with the on/off times recorded by a trained educator. Mid-day naps were removed from the data and were not classified as non-wear time.

### 2.4. Data Analysis and Statistical Methods

Questionnaire data and accelerometer data were analyzed in SPSS Statistics version 26 (IBM SPSS, Chicago, IL, USA) with statistical significance set to *p* < 0.05. Cohen’s d was used to determine the effect size of significant differences and was interpreted as 0.2 = small effect, 0.5 = moderate effect, and 0.8 = large effect.

Descriptive statistics for participant demographics and physical activity intensities were calculated. An independent t-test was employed to measure the difference between the CHEERS scores (and all subscales—FS, HEE, HEP, and PAE) and MEQ scores (and all subscales—disinhibition, awareness, external cues, emotional response, and distraction) for the Y-19 and Y-20 groups, respectively. The ability of CHEERS to detect differences between groups may add to its construct validity. A Pearson r was employed to determine the relationship between the CHEERS and the MEQ. Although these two tools have a different purpose, it is postulated that there are some overlaps in the underlying constructs being measured in each. The correlation may also act as a concurrent validation of the CHEERS tool.

Data were checked for normality using Shapiro–Wilk and Q–Q plots. Independent samples t-tests were conducted to examine differences between groups in terms of percentage of the day spent in sedentary, in LPA, in MVPA, and TPA. Levene’s test revealed unequal variances for outcome measures of the following activity intensities in both min per hour and percentage of the day spent in sedentary (F = 6.73, *p* = 0.01), LPA (F = 10.62, *p* = 0.001), and TPA (F = 6.73, *p* = 0.01). Degrees of freedom were adjusted from 236 to 229, 234, and 229, respectively, and equal-variances-not-assumed test statistics are reported below.

## 3. Results

### 3.1. Descriptive Analysis

Forty-five licenced ECEC centers in 2019 and thirty-three centers in 2020, from two metropolitan and two mid-sized cities across Alberta, Canada, were eligible to participate in the study. A total of thirty-four centers across the two years were included in the final sample with a balance of both large urban and medium population centers in both years (Table 1). Information on education level for one of the educators was missing in Y-20.

From thirty-four participating ECEC centers, 361 eligible children participated in the accelerometer data collection. Of the 361 eligible children, 238 (Y-19: *n* = 95; Y-20: *n* = 143) aged 36 to 71 months had valid accelerometer data and were included in the analysis (94 children were excluded due to invalid wear time, and 29 were excluded because their age was less than 36 months at the time of data collection). Descriptive participant characteristics by year are provided in Table 2. The average child age was just over 50 months and approximately half of the children were female for both groups. No significant difference was observed in accelerometer total wear time in Y-19 (43.2 h analyzed) compared to Y-20 (41.0 h analyzed). Average wear time for the Y-19 group was 7.36 h and 6.85 h in the Y-20 group. Similarly, the number of valid days was stable across the two groups. BMI for age was statistically significantly greater in the pandemic Y-20 group (16.4 ± 2.6) compared to the non-pandemic Y-19 group (15.8 ± 1.6) at *p* = 0.047.

### 3.2. Questionnaire Results

An independent t-test was employed to measure differences between eating and activity environments in 2019 (pre-pandemic *n* = 145) and 2020 (pandemic *n* = 95). There was no significant impact on the overall CHEERS score between years. However, the healthy eating environment subscale was greater in the Y-20 group (5.80 ± 0.62) compared to the Y-19 group (5.97 ± 0.52), (*p* = 0.02 (two-tailed)). The magnitude of the difference in the means was small (*d* = 0.297). Descriptive data and all relevant comparative analysis for all CHEERS subscales are listed in Table 3.

An independent t-test was employed to measure differences between educators’ responses to mindful eating practices in 2019 (pre-pandemic *n* = 39) and 2020 (pandemic *n* = 44). The overall MEQ scores between years was not significantly different; however, the subscale ‘external cues’ was marginally significant in the Y-20 group (2.59 ± 0.49) compared to the Y-19 group (2.82 ± 0.61), (*p* = 0.056 (two-tailed)). Descriptive data and all relevant comparative analyses for the MEQ subscales are listed in Table 4.

### 3.3. Concurrent Validation of CHEERS

Pearson’s *r* correlation coefficient was employed to evaluate the relationship between the CHEERS and the MEQ. There was a statistically significant correlation between the two survey tools: *r* = 0.20, *p* = 0.002.

### 3.4. Accelerometer Assessed Physical Activity of Preschool Children

The sedentary time, LPA, MVPA, total physical activity (TPA), and step count during childcare hours are shown in Table 5. The sedentary time significantly (*p* < 0.01) decreased between Y-19 (44.2 min/h or 73.4% of wear time) and Y-20 (42.4 min/h or 70.7% of wear time) with a Cohen’s *d* value of 0.363. The LPA was unchanged between years whereas the MVPA was significantly greater in Y-20 (9.3 min/h or 15.5% of wear time) compared to Y-19 (7.9 min/h or 13.2% of wear time) (*p* < 0.01, *d* = 0.434). Overall, preschoolers engaged in more movement as total physical activity in the Y-20 COVID-19 condition compared to similar aged preschoolers in the Y-19 group (*p* < 0.01, *d* = 0.360).

The hourly step count of preschoolers during childcare hours in the COVID-19 condition was significantly greater (800 ± 189 steps/h) than preschoolers in the Y-19 group (649 ± 186 steps/h) (*p* < 0.001, *d* = 0.802). The overall daily step count was significantly greater in the Y-20 group (5507 ± 1748 steps/d) compared to the Y-19 pre-COVID-19 group (4848 ± 1772 steps/d) (*p* = 0.005, *d* = 0.340).

## 4. Discussion

The healthy-eating environment CHEERS subscale demonstrated a statistically significant difference between a COVID-19-year compared to a non-COVID-19 year. In addition, the sedentary time was lower while the MVPA and step count were higher in the context of COVID-19. Government guidelines for remaining open in the fall of 2020 were focused on reducing transmission of the virus and a key message was physical distancing. Overall, increased ECEC center attention regarding eating environments and increased child movement could be attributed to a change in policy as actioned through the government-mandated COVID-19 guidelines.

### 4.1. COVID-19 and Nutrition Environments in ECEC Setting

Because eligibility requirements for childcare centers were matched for similar geographic locations, city center sizes, auspices (profit/not-for-profit), and preschooler-care characteristics between recruitment years, baseline data for these groups were expected to be similar [20,21]. However, the results in this study indicate that the healthy eating environment subscale of CHEERS was greater in the 2020 group. The worldwide emergence of COVID-19 changed aspects of public life, altering the normal operations throughout the world and, by association, the ECEC environments. The Government of Alberta reopening guidelines for licensed childcare facilities implemented a variety of measures to reduce the potential for transmission of the virus that impacted the nutrition environment. ECEC centers serving meals to children follow national food guide recommendations, serve children the same meal, and have the ability to provide family-style meal service. Food-related COVID-19 government guidelines in 2020 reduced educator-to-child ratios, prohibited family-style meal services, increased sanitization requirements, and recommended physical distancing during meals in childcare facilities [3]. These guidelines drew attention to critical health practices and eating environments. In Alberta, changes in food provision for ECEC centers early in COVID-19 reopening contexts resulted in many parents providing food as many centers did not initially bring back cooks into programs [33]. In a previous qualitative study, educators noted a change in the quality and/or quantity of meals and snacks supplied by families [33]. Financial challenges experienced by families were also identified as a possible reason for reduced nutritional quality or quantity as many families faced financial challenges, which possibly resulted in food insecurity [34]. Financial challenges were also experienced by ECEC programs that struggled to continue center-provided food service for children in care [35]. By the fall of 2020, the ECEC programs included in this study had returned to providing food for children in their care, which may account for the similarity in the food served subscale scores between the 2019 and 2020 groups. These contexts may have raised awareness of the pivotal role ECEC programs and educators play in healthy eating environments and prompted increased focus and action, as suggested by the CHEERS subscale scores in 2020.

### 4.2. COVID-19 and the Physical Activity in the ECEC Setting

The second key finding of this study was that preschoolers (3–5 yrs) spent more time in overall physical activity (TPA) and moderate-to-vigorous activity (MVPA), and took more steps during childcare hours in the COVID-19 context compared to the previous year. In contrast, a national Canadian study of children aged 5–11 years found that physical activity levels were lower during COVID-19 in spring 2020 and were associated with less outside time [36]. The differences with our findings of increased physical activity may be related to the government guidelines on structured care settings that were connected to their COVID-19-mitigation strategy, as well as to increased outdoor time. The guidelines focused on physical distancing and the sanitizing of shared equipment between cohorts. Further, it was recommended that centers use alternatives to licensed outdoor spaces such as walks, supervised play in parks, and safe open spaces [3]. These governmental guidance recommendations created a form of policy that informed the ECEC program’s daily practice. Physical activity policies in early childhood education settings may play a role in supporting children’s physical activity levels. In Canada, policy standards (accreditation) were demonstrated to have a positive impact in terms of modestly increasing physical activity and decreasing sedentary behaviors in toddlers, suggesting the responsiveness of childcare programs to policy implementation [37]. However, the policies themselves provide no guarantee of increasing physical activity [38]. The positive effect of the COVID-19 governmental policy impact observed in the current study might be explained through the sociopolitical context and authority from which it was implemented.

Within a critical ecology framework, educators interact within the layers of systems that influence their professional practice [39]. Educators actively interact in multiple microsystems, whereas exosystems and macrosystems represent layers where decisions impact educators. Physical distancing and alternative activity recommendations were powerful messages delivered to ECEC programs. These messages filtered through mesosystems and, when merged with educators’ physical activity professional practice perspectives, may have created the motivation to action physical activity policies within the childcare program. In a recent qualitative study on COVID-19 experiences among early childhood educators, they expressed that spending more time outdoors with children eased the tension associated with trying to enforce physical distancing in their classroom [33]. Time spent outdoors has been found to be a strong, consistent predictor of children’s physical activity [40,41]. In Canada, preschoolers were ten times more active in outdoor environments than indoor environments [42]. The increased physical activity identified in this study as a result of the COVID-19 circumstances may be the outcome of increased time spent outdoors. This appears to be an accessible strategy that could have broad population-level benefits with sufficient motivation to implement the policy.

The results of this study demonstrate that a sample of preschoolers throughout the province of Alberta spent 16 and 17.9 min/h engaging in any-intensity physical activity (TPA) in 2019 and 2020, respectively. This falls within the range found in similar studies looking at preschool-aged children TPA during care, with results ranging from 14.1 to 21.9 min/h [43,44]. Our results are also consistent with Canadian childcare TPA findings that range from 15.6 to 19.5 min/h [45]. In preschool children, 15 min/h of TPA while children are in care is the Institute of Medicine (IOM) recommendation standard to ensure that toddlers and preschool children have appropriate opportunities to be physically active throughout the day [46]. On average, preschoolers in the current study achieved the IOM minimum recommendations in both the pre-COVID-19 and COVID-19 context. The Canadian 24-h movement guidelines for preschoolers (3–4 years) recommend 10–13 h of quality sleep and 180 min of TPA, of which at least 60 min should be energetic play throughout the child’s day [18]. On average, Canadian children participating in early learning and childcare spend approximately 6 h/day in care [8]. In light of these recommendations, children achieved approximately 96 min/d (53%) and 107 min/d (60%) of daily recommended activity during care hours in 2019 and 2020, respectively. In the current study, children wore accelerometers only during childcare hours. If children in care spend approximately two-thirds of their day in care, adjusting the recommendations for two-thirds of the 180 min/d of activity would estimate a 120 min/d minimum activity target for childcare centers. In terms of these targets, the children achieved 96 min/d (80%) and 107 min/d (90%) of a 120 min/d target in 2019 and 2020, respectively.

In the current study, the children spent 7.9 min/h in MVPA prior to COVID-19 conditions. This aligns with similar studies, the results of which show a range from 5.5 to 14.1 min/h MVPA for preschoolers attending care [38,44]. A systematic review and meta-analysis reported that preschool-aged children in structured care settings spend approximately 7.9 min/hour in MVPA, as measured with ActiGraph accelerometers based on Pate cut-points [47]. Canadian guidelines of at least 60 min of energetic play would equate to approximately 10 min/h of MVPA throughout care hour. Children in the current study achieved an average of 7.9 min/h (79%) in 2019. In the COVID-19 context, children achieved higher hourly MVPA (9.3 min/h) and reached 93% of the Canadian energetic play recommendations.

### 4.3. COVID-19 and Sedentary Behaviours of Preschoolers in ECEC Settings

Preschoolers (3–5 yrs) spent statistically fewer sedentary min (42.4 min/h) in the 2020 COVID-19 context compared to pre COVID-19 (44.2 min/h). While these estimates indicate a large proportion of time spent overall in sedentary activities, they are consistent with findings on children in care, which range from 31.1 to 45.9 min/h in sedentary behaviour [38,44,47]. Our results are also consistent with Canadian childcare sedentary findings, which range from 40.7 to 44.5 min/h [45].

### 4.4. COVID-19 and Preschooler’s BMI

BMI for age was slightly higher in the 2020 COVID-19 context preschool group (16.4 ± 2.6 kg/m^2^) compared to the 2019 group (15.8 ± 1.6 kg/m^2^). This is consistent with reports worldwide that identify an increase in BMI as a result of COVID-19 in preschool-aged children [48,49,50], youth 6–12 years [51], and adults [52]. However, the increased BMI appears to conflict with the increased healthy eating CHEERS subscale, MVPA levels, and step count in the 2020 COVID-19 context preschoolers. However, at the time of the 2020 data collection, children were returning to childcare after spending extended time in the family home during the first worldwide lockdown [2]. In a Canadian study, parents reported that their eating habits at home during the COVID-19 lockdown included eating more snack foods and engaging in less physical activity [36]. In addition, educators noted that when children returned to childcare centers in the summer of 2020, food supplied by parents initially included more sweets and treats than when centers supplied meals [33]. The slight increase in preschooler BMI might be representative of home life during the pandemic lockdown.

### 4.5. CHEERS Construct and Concurrent Validity

The CHEERS survey measures gaps, weaknesses, and strengths of ECEC center-based nutrition and physical activity environments. The mindful eating questionnaire (MEQ) provides a non-judgmental awareness of physical and emotional sensations with eating. CHEERS and MEQ measure overlapping constructs related to healthy eating constructs. The overall MEQ score was significantly associated with CHEERS overall score (*r* = 0.20, *p* = 0.02). The alignment between the MEQ awareness subdomain with the CHEERS healthy eating environment and program planning domains provides evidence of concurrent validity for the CHEERS audit tool.

The COSMIN guidelines describe responsiveness of an instrument as the ability to detect change over time or between groups in the construct being measured [53]. In the current study, a small increase in the CHEERS healthy eating environment subscale was observed in the 2020 sample. Sampling characteristics were consistent between the two years; however, the difference detected by the CHEERS tool in light of the pandemic conditions provides evidence of responsiveness for the CHEERS tool. Taken together, both these findings provide further evidence of overall validity and reliability of the CHEERS tool.

### 4.6. Limitations and Future Directions

There are some limitations in the present study that must be considered when interpreting the data. Sampling for this study was limited to the Province of Alberta and although the ECECs were selected randomly, the scope of sampling will still limit the study’s generalizability. Secondly, the first cohort recruited in 2019 were part of a larger study that participated in an intervention and therefore could not be used again to evaluate the COVID year due to that intervention. A second convenience sample was recruited in 2020 and their baseline scores, without any intervention, were used as a comparison to the 2019 cohort. None of these comparisons were planned a priori, but rather, the circumstances of COVID facilitated the comparisons, which also contributed to unequal sample sizes between the 2019 and 2020 groups. Thirdly, nutrition in ECEC centers was self-assessed, which introduced self-report bias. Lastly, accelerometers cannot detect postural changes. This inability may have resulted in some measurement error of sedentary behaviour. Despite these limitations, these results contribute to a better understanding of the relationship between COVID-19 and ECEC eating and activity environments.

Our findings have implications for practice and research. The first implication pertains to the impact of policy on educator practice. While there is awareness of good health practice among educators, policy drives educator decision making and action in terms of practice. Policies for nutrition and physical activity should be comprehensive, clear, evidence-based, measurable, and evaluated to facilitate best-practice implementation in the ECEC classroom. The second implication is the significant perturbation in the environment, caused by pandemic physical distancing and heavy cleaning protocols, which altered movement patterns and increased outdoor time in ECEC centers. Evidence suggests that while this altered normal is still fresh, now is a critical time to support the emergent practice changes that resulted in the increased physical activity of young children.

Educator health knowledge and practices play an important role in the development of a child’s relationship with food and activity for a lifetime. Future research should investigate approaches to support educators and ECEC centers to (1) further develop their understanding and knowledge of physical activity and physical literacy in the context of the early childhood education curriculum, (2) further explore knowledge and implementation strategies to increase the professional practice uptake of healthy eating evidence-based best practice, and (3) increase their use of evaluation tools to measure and track progress.

## 5. Conclusions

The purpose of this study was to assess the impact of COVID-19 government-mandated guidelines on healthy eating and physical activity environments in ECEC settings from baseline measurements of a larger multi-year health and wellness initiative. ECEC programs were selected to match center characteristics. The healthy eating environment CHEERS subscale was greater for childcare centers in the COVID-19 context but the MEQ was stable between years. The responsiveness of the subscales to pandemic conditions and concurrent validity between CHEERS and MEQ provides evidence on the psychometric properties of the CHEERS tool as an ECEC environment auditing tool. Children aged 36–71 months from these centers were more active in the COVID-19 context compared to similar childcare programs during the previous non-COVID-19 year. Children took more steps, had significantly lower sedentary time, and met 60% of the daily physical activity recommendations within childcare hours. Guidelines to keep children physically distanced and reduced equipment sanitization requirements while out in the community may be key factors. These findings suggest that policy at the ECEC level may have a substantial impact in terms of supporting educators in implementing best practices. Considering that children in care tended to not fully meet physical activity recommendations, we suggest that a focus on outdoor time continue to be promoted.

## Figures and Tables

**Table 1 nutrients-13-04247-t001:** ECEC center and educator characteristics before and during the COVID-19 pandemic.

	Y-19	Y-20
Centers	19	15
Not-for-profit	12	8
For-profit	7	7
Geographic location		
Large urban population center	17	12
Medium population center	2	3
Educators	39	44
Sex (% female)	100%	97.7%
Age (years)	38.1 ± 12.6	42.3 ± 12.2
Education		
CDA ^1^	3	6
CDW ^2^	5	4
CDS ^3^	15	26
University degree	16	7

^1^ Child development assistant; ^2^ child development worker; ^3^ child development supervisor.

**Table 2 nutrients-13-04247-t002:** Characteristics of preschoolers in the sample.

	Y-19	Y-20
Preschoolers	143	95
Sex (% female)	51.1	53.7
Age (months)	51.5 ± 8.1	50.3 ± 7.9
BMI for age	15.8 ± 1.6	16.4 ± 2.6 *
Accelerometer wear time (h)	43.2 ± 10.6	41.0 ± 12.4
Number of valid days (d)	5.9 ± 1.0	5.9 ± 1.1

* Significantly different at *p* < 0.05.

**Table 3 nutrients-13-04247-t003:** CHEERS and subscale scores for ECEC environments before and during the COVID-19 pandemic.

	Y-19	Y-20	*p*-Value	Cohen’s d
CHEERS score	5.40 ± 0.68	5.49 ± 0.63	0.34	0.137
food served	5.82 ± 0.62	5.89 ± 0.52	0.35	0.122
healthy eating environment	5.80 ± 0.62	5.97 ± 0.52 *	0.02	0.297
healthy eating program planning	4.35 ± 0.98	4.38 ± 1.12	0.79	0.029
physical activity environment	5.65 ± 0.76	5.70 ± 0.62	0.57	0.072

* Significantly different at *p* < 0.05.

**Table 4 nutrients-13-04247-t004:** MEQ and subscale scores for educators working in ECEC-licensed centers before and during the COVID-19 pandemic.

	Y-19	Y-20	*p*-Value	Cohen’s d
MEQ score	3.03 ± 0.27	3.02 ± 0.27	0.87	0.033
disinhibition	3.28 ± 0.48	3.30 ± 0.53	0.90	0.026
awareness	2.99 ± 0.46	2.90 ± 0.53	0.40	0.187
external cues	2.82 ± 0.61	2.59 ± 0.49	0.06	0.422
emotion	3.07 ± 0.71	3.26 ± 0.56	0.35	0.298
distraction	3.03 ± 0.55	3.11 ± 0.53	0.51	0.146

**Table 5 nutrients-13-04247-t005:** Physical activity in preschool children during time spent in childcare and as a percent of time spent in childcare before and during the COVID-19 pandemic.

	Y-19	Y-20	*p*-Value	Cohen’s d
Sedentary time				
Min/h	44.2 ± 4.9	42.4 ± 3.9 *	0.005	0.363
Percentage of wear time	73.4 ± 8.2	70.7 ± 6.5 *	0.005	0.363
LPA (min/h)				
Min/h	8.0 ± 1.9	8.3 ± 1.4	0.223	0.155
Percentage of wear time	13.6 ± 3.2	13.8 ± 2.3	0.223	0.157
MVPA (min/h)				
Min/h	7.9 ± 3.2	9.3 ± 3.0 *	0.001	0.434
Percentage of wear time	13.2 ± 5.3	15.5 ± 5.0 *	0.001	0.435
TPA				
Min/h	16.0 ± 4.9	17.6 ± 3.9 *	0.005	0.363
Percentage of wear time	26.6 ± 8.2	29.3 ± 6.5 *	0.005	0.327
Steps Count				
Steps/h(steps/h)	649 ± 186	800 ± 189 **	0.000	0.802
Steps/d	4848 ± 1772	5507 ± 1748 *	0.005	0.374

* Significantly different at *p* < 0.01; ** significantly different at *p* < 0.001.

## Data Availability

The datasets generated during and/or analyzed during the current study are not publicly available due to the terms of participant consent.

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
