# Peer review of "The Impact of COVID-19 on Eating Environments and Activity in Early Childhood Education and Care in Alberta, Canada: A Cross-Sectional Study"

_nutrients, 2021, doi:10.3390/nu13124247_

Round 1

Reviewer 1 Report

Dear authors, 

First of all, I would like to congratulate the authors for their efforts in carrying out this research. Overall, the purpose of the study is focused on the impact of COVID-19 guidelines on physical activity and healthy eating in ECEC contexts. The topic is important and relevant, and the measures are robust (e.g., the use of accelerometry). However, I have some questions and concerns about the structure and writing of the article.

For this reason, I offer the below suggestions that could be addressed to improve the quality of the paper before publication:

INTRODUCTION: The introduction is well-written, provides a concise background; the references are sufficiently up-to-date, and the goal of the manuscript is clear.

MATERIALS AND METHODS:

In the Participants and Design section, some questions emerged:

  1. This is a cross-sectional study involving 34 ECEC centers.

However, it was not clear how many ECEC educators and staff were included in the study. Can the authors add this information (see table 1 suggestion)?

  1. In the line section childcare center and educators (2.1.1), the authors explain how the recruitment of the ECEC was made. However, it was also not clear if the sample from Y19 (e.g., centers and participants) were totally different from the sample of Y20. And, if this procedure, was equal in both moments (recruiting different centers and, therefore, different participants).

Please clarify this information.

RESULTS:

  1. Table 1: a line with a total sample of the educators in each Year could be more informative.

  1. Line 238: the whole paragraph is repeated. Please, correct.

DISCUSSION: Overall, this section need improvement to facilitate the reading and understanding of the main results. For example:

  1. Ln 416: regarding the preschooler’s BMI (subsection 4.4.), the authors referred that the BMI for age was lower in the sample of Y19. This result is confusing. How is this result consistent with the previous results? For instance, previously, the authors showed that the sample Y20 was more active in the COVID 19 context, and the healthy eating environment was higher in the 2020 group. Is this result relevant enough to discuss separately in the discussion? Please, clarify.

  1. ln 444: To facilitate the reading of the discussion, I suggest adding a 4.6 section called limitations and future directions.

By adding this subsection, a detailed description of future directions could be developed, enriching the manuscript conclusion and guiding future work in more detail.

Congratulations for your work. 

Author Response

We would like to extend our gratitude to the reviewers for taking the time to review the study. We appreciate the feedback and have outlined the following modifications to prepare the manuscript suitable for publication.

General Comments

Reviewer #1 - First of all, I would like to congratulate the authors for their efforts in carrying out this research. Overall, the purpose of the study is focused on the impact of COVID-19 guidelines on physical activity and healthy eating in ECEC contexts. The topic is important and relevant, and the measures are robust (e.g., the use of accelerometry). However, I have some questions and concerns about the structure and writing of the article.

For this reason, I offer the below suggestions that could be addressed to improve the quality of the paper before publication.

Author Response - Thank you for your comment. We have carefully edited with this in mind.

INTRODUCTION

Reviewer #1 - The introduction is well-written, provides a concise background; the references are sufficiently up-to-date, and the goal of the manuscript is clear.

Author Response – thank you we appreciate your comments

MATERIALS AND METHODS

Reviewer #1 - This is a cross-sectional study involving 34 ECEC centers. However, it was not clear how many ECEC educators and staff were included in the study. Can the authors add this information (see table 1 suggestion)?

Author Response - Thank you. This has been changed as suggested.

Reviewer #1 - In the line section childcare center and educators (2.1.1), the authors explain how the recruitment of the ECEC was made. However, it was also not clear if the sample from Y19 (e.g., centers and participants) were totally different from the sample of Y20. And, if this procedure, was equal in both moments (recruiting different centers and, therefore, different participants).

Please clarify this information.

Author Response – We agree this needs clarity. We have added a sentence to clarify this information

“In 2020, a new cohort of centers were contacted to participate in the study following the same protocol.”

RESULTS

Reviewer #1 - Table 1: a line with a total sample of the educators in each Year could be more informative.

Author Response - Thank you. This has been changed as suggested in Table 1.

Reviewer #1 - Line 238: the whole paragraph is repeated. Please, correct.

Author Response – the paragraph has been deleted.

DISCUSSION

Reviewer #1 - Ln 416: regarding the preschooler’s BMI (subsection 4.4.), the authors referred that the BMI for age was lower in the sample of Y19. This result is confusing. How is this result consistent with the previous results? For instance, previously, the authors showed that the sample Y20 was more active in the COVID 19 context, and the healthy eating environment was higher in the 2020 group. Is this result relevant enough to discuss separately in the discussion? Please, clarify.

Author Response – this increase in BMI as a result of COVID lockdown is aligned with the literature (Weaver et al., 2021; Wen et al., 2021; and Woolford et al., 2021) so it is something to note. We have contextualized the BMI increase “The slight increase in preschooler BMI might be representative of home life during lockdown.

Reviewer #1 - ln 444: To facilitate the reading of the discussion, I suggest adding a 4.6 section called limitations and future directions.

By adding this subsection, a detailed description of future directions could be developed, enriching the manuscript conclusion and guiding future work in more detail.

Author Response – Thank you for your suggestions. We added heading 4.6 and provided more detail for future research directions.

Reviewer 2 Report

I found it interesting to read this article. Method and results are well written and presented in detail. Have a few minor suggestions to improve the paper.

Lines 13-14: Revise the sentence

Lines 299-300: sub-scale of CHEERS tool?

Lines 309-313: Move to Methods

Lines: 331-333: Revise the sentence

Materials and methods: Power and Sample Size calculation? 

Data analysis and statistical methods: Adjustment of P-value for multiple comparisons?

Discussion: add justification for higher prevalence of BMI for age in Y-20, though having higher PA

Unequal sample size: Y-19 and Y-20, better to add in the limitations

Add P values in the table

Better to run a regression model by adjusting confounding variables such as BMI

Author Response

We would like to extend our gratitude to the reviewers for taking time to review the study. We appreciate the feedback and have outlined the following modifications to prepare the manuscript suitable for publication.

Reviewer #2 - I found it interesting to read this article. Method and results are well written and presented in detail. Have a few minor suggestions to improve the paper.

Author Response - Thank you for your comment. We have carefully edited with this in mind.

Reviewer #2 - Lines 13-14: Revise the sentence

Author Response – we revised the sentence for clarity. “This cross-sectional study involved data collection pre-COVID (2019) and during COVID (2020) from 34 ECEC centers in Alberta, Canada (n = 361 preschoolers; n = 83 educators).”

Reviewer #2 - Lines 299-300: sub-scale of CHEERS tool?

Author Response – we revised the sentence to clarify the healthy eating environment as a CHEERS subscale score

Reviewer #2 - Lines 309-313: Move to Methods

Author Response – we have moved the eligibility requirement to the methods section. We now refer to the idea of matched eligibility to provide context to the expectation that the groups would be similar at baseline.

Reviewer #2 - Lines: 331-333: Revise the sentence

Author Response – we have revised the sentence as suggested. “Financial challenges were also experienced by ECEC programs who struggled to continue center provided food service for children in care [35].”

Reviewer #2 - Materials and methods: Power and Sample Size calculation? 

Author Response – This was an observational study and as a result, there were no a priori power calculations. However, saying that, we felt that a suitable alternative that would speak to the “power” of the study is the effect size. We do not feel it would be appropriate to put in a power calculation after the fact. Because the study is observational and a convenience sample, we have reported the statistical significance and the effect size, which also speaks to the purpose of sample size directly, in order to address this comment.

Reviewer #2 - Data analysis and statistical methods: Adjustment of P-value for multiple comparisons?

Author Response – because statistical tests were completed on between-group differences Y19 vs Y20 only (MEQ scores and subscales & CHEERS scores and subscales), no adjustment of p-value was employed.

Reviewer #2 - add justification for higher prevalence of BMI for age in Y-20, though having higher PA

Author Response – the increase in BMI for preschoolers in the Y-20 group as a result of COVID lockdown is aligned with the literature (Weaver et al., 2021; Wen et al., 2021; and Woolford et al., 2021). We have added a justification of why this BMI increase might have occurred even in the presence of increased MVPA/step count. The BMI increase is representative of extended at-home time during lockdown which also aligns with what has been reported in the literature. The increased step count is current in child care centers post lockdown in response to physical distancing.

Reviewer #2 - Unequal sample size: Y-19 and Y-20, better to add in the limitations

Author Response – a separate section was added for limitations (4.6) and the unequal sample size between cohorts was included in the limitations.

Reviewer #2 - Add P values in the table

Author Response – p values have been added to results Table 3 – 5.

Reviewer #2 - Better to run a regression model by adjusting confounding variables such as BMI

Author Response – We feel like this type of analysis could overly complicate our results and our attempt to create a simple message considering the immense amount of data that we are already reporting on in this paper. The problem with completing a regression analysis for BMI, a measure with the children, and using those data in the regression model with the CHEERS or MEQ is that the respondent of the latter two tools is the early childhood educator, not the child. In other words, the model would mix data collected from the child with data collected from the early childhood educator. The interpretation of any results would be tainted because of the mixture of data sources since not all children were at the same childcare center skewing whatever results we would get from this model. As a result, we chose to aggregate the children’s data to evaluate BMI, thus comparing children to children. The comparison of CHEERS and MEQ permit comparison of educator to educator.